# Chiral $p$-wave superconductivity in twisted bilayer graphene from dynamical mean field theory

**B. Pahlevanzadeh[1,2], P. Sahebsara[1] and David Sénéchal[2⋆]**

**1** Department of Physics, Isfahan University of Technology, Isfahan, Iran
**2** Département de physique and Institut quantique, Université de Sherbrooke, Sherbrooke, Québec, Canada J1K 2R1

⋆ david.senechal@usherbrooke.ca

Supplementary information: https://osf.io/sd4bk/

## Abstract

We apply cluster dynamical mean field theory with an exact-diagonalization impurity solver to a Hubbard model for magic-angle twisted bilayer graphene, built on the tight-binding model proposed by Kang and Vafek [1], which applies to the magic angle 1.30°. We find that triplet superconductivity with $p + ip$ symmetry is stabilized by CDMFT, as well as a subdominant singlet $d + id$ state. A minimum of the order parameter exists close to quarter-filling and three-quarter filling, as observed in experiments.



## 1 Introduction

Twisted bilayer graphene (TBG) consists of two layers of graphene deposited on top of each other with a slight rotation, or twist. At commensurate twist angles, the bilayer forms a moiré

pattern with a period that depends closely on the twist angle. It has been predicted that for some "magic angles", the resulting band structure has a few relatively flat bands at low energy, separated from the rest, thus forming an effective strongly interacting electronic system [2–4]. The physical realization of this occurred in 2018 when Cao *et al.* observed Mott behavior in quarter-filled TBG (filling is understood here in terms of the four low-energy bands) at some magic angles [5] and detected superconductivity just away from that filling [6]. Superconductivity was also found at larger twist angles by applying pressure [7]. These discoveries have renewed theoretical research on this system, with the goal of understanding the origin of superconductivity in TBG [8–17]. Some authors have found triplet superconductivity to be dominant [9,13], others predict singlet superconductivity, specifically of the $d + id$ type [8, 14, 16, 17]. The great variety of effective models and methods used complicates the comparison between these works.

The difficulty here is two-fold: (i) to construct a model Hamiltonian that can reasonably represent this very complex system and (ii) to predict correctly, within that model, whether superconductivity arises, and if so, with what characteristics: singlet or triplet, order parameter symmetry, etc.

Since magic angle TBG is a strongly correlated system, the natural course of study is to set up an effective low-energy Hamiltonian in the Wannier basis, as opposed to the Bloch basis [1, 18–20]. Since the moiré pattern of TBG forms a triangular lattice, it was initially thought that the effective Hamiltonian would be defined on that lattice, and indeed it was shown that the electron density associated with the low-energy bands is peaked around its sites. However, it was then shown that no Wannier basis satisfying the minimal symmetry requirements could be constructed on a triangular lattice; on the contrary, the Wannier states have to be defined on the plaquettes of a triangular lattice, which form a graphene-like (hexagonal) lattice.

We adopt as a starting point the model proposed by Kang and Vafek [1], itself based on the microscopic analysis of Moon and Koshino [19]. We then simply add a Hubbard $U$, local to each of the four Wannier states per unit cell, and apply cluster dynamical mean field theory (CDMFT) to this interacting model in order to probe specific superconducting states. We find that a superconducting state indeed exists around quarter filling and three-quarter filling and that it is a triplet state with $p + ip$ symmetry, while a subdominant, singlet $d + id$ solution also exists. This is the main conclusion of this work.

## 2   Low-energy model

There have been a few proposals for an effective tight-binding Hamiltonian describing the low-energy bands of TBG [1, 18–20]. We adopt in this work the model described in Ref. [1] and inspired by Ref. [19]. It is based on four Wannier orbitals per unit cell, with maximal symmetry, on an effective honeycomb lattice and is appropriate for a twist angle $\theta = 1.30°$.

It is customary to derive effective models for TBG directly from continuum models. In that framework a valley symmetry emerges and the model is endowed with a fragile topology. It can be shown that in a model with nontrivial topology, time-reversal symmetry (TRS) cannot be represented simply by a set of localized Wannier states: its action is not strictly local [21]. However, as shown in [22], the error committed by using a localized Wannier basis is exponentially small. Since we are going to truncate the hopping matrix to a few terms and introduce strong interactions that would likely destroy any existing topology, this issue should not be of concern here.

Fig. 1 offers a schematic view of the orbitals $w_1$ and $w_3$. Orbitals $w_2 = w_1^*$ and $w_4 = w_3^*$ are not shown. Ref. [1] computes a large number of hopping integrals, of which we will only retain the largest, as listed in Table 1. The notation used is that of Ref. [1].

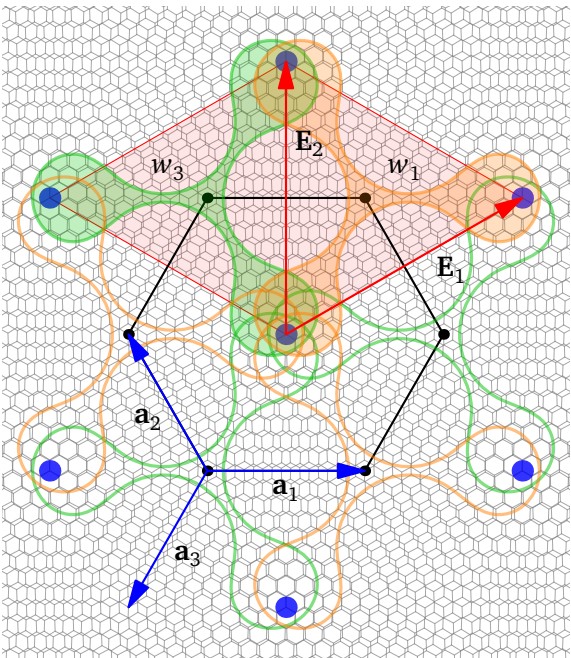

Figure 1: Schematic representation of the Wannier functions $w_1 = w_2^*$ (orange) and $w_3 = w_4^*$ (green) on which our model Hamiltonian is built. The charge is maximal at the AA superposition points (blue circles) forming a triangular lattice. The Wannier functions are centered on the triangular plaquettes that form a graphene-like lattice (black dots), whose unit cell is shaded in red. The underlying moiré pattern illustrated corresponds to $(m, n) = (9, 8)$, but the functions used in this work correspond to $(m, n) = (26, 25)$. The basis vectors $\mathbf{E}_{1,2}$ of the moiré lattice are shown (they are also basis vectors of the graphene-like lattice of Wannier functions), as well as the elementary nearest-neighbor vectors $\mathbf{a}_{1,2,3}$.

Remarkably, the most important hopping terms are between $w_1$ and $w_4$ (and between $w_2$ and $w_3$), i.e., between graphene sublattices. It therefore makes sense physically to picture the system as made of two *layers* and to assign $w_1$ and $w_4$ to the first layer, whereas $w_2$ and $w_3$ are assigned to the second layer. The rather small $t_{13}[0,0]$ hopping (and its equivalents) is the only term that couples the two layers. The concept of layer is useful when visualizing the model in space and when arranging local clusters of sites in CDMFT, since it is preferable to have the more important hopping terms within a cluster; it is merely a book-keeping device. The drawing next to Table 1 illustrates the range and multiplicity of the intra-layer hopping terms retained.

To this tight-binding model we will add a local interaction term $U$. This is a rather approximate description of the interactions in this system, but has the merit of simplicity and tractability in the context of dynamical mean field theory. A more refined description of the interactions would not only contain extended interactions (see, e.g., [23,24]) but would include terms not of the density-density form [25]. We will defer the study of extended interactions to future work. The values of $U$ in our calculations range from 0.5 meV to 5 meV. Fig. 3d of Ref. [5] leads us to expect a wide range of values of $U$ depending on twist angle, and a rather large $U \sim 20$ meV at an angle of 1.30°. However, Ref. [11] predicts a value $U \sim 5$ meV for this angle and the range of $U$ values predicted in Fig. 9 of Ref. [26] is largely compatible with the range we have selected.

The model is invariant under a rotation $C_3$ by $2\pi/3$ about the AA site, and under a $\pi$-rotation $C_2'$ about an axis in the plane of the bilayer (the vertical axis on Fig. 1). These trans-

Table 1: Hopping amplitudes used in this work. They are the most important amplitudes computed in Ref. [1]. Here $\omega = e^{2\pi i/3}$ and the vector $[a, b]$ following the symbol represents the bond vectors in the $(\mathbf{E}_1, \mathbf{E}_2)$ basis shown on Fig. 1. Note that $t_{23} = t_{14}^*$ and $t_{24} = t_{13}^*$. On the right: schematic view of the hopping terms $t_{14}$ within a given layer (the unit cell is the blue shaded area). Lines 2, 3, and 4 of the table correspond to the red, blue and green links, respectively. Dashed and full lines are for $t_{14}$ and $t_{23}$, respectively.

| symbol | value (meV) |
|---|---|
| $t_{13}[0,0] = \omega t_{13}[1,-1] = \omega^* t_{13}[1,0]$ | $-0.011$ |
| • $t_{14}[0,0] = t_{14}[1,0] = t_{14}[1,-1]$ | $0.0177 + 0.291i$ |
| • $t_{14}[2,-1] = t_{14}[0,1] = t_{14}[0,-1]$ | $-0.1141 - 0.3479i$ |
| • $t_{14}[-1,0] = t_{14}[-1,1] = t_{14}[1,-2]$ | |
| $= t_{14}[1,1] = t_{14}[2,-2] = t_{14}[2,0]$ | $0.0464 - 0.0831i$ |

formations generate the point group $D_3$ and affect the Wannier orbitals as follows [1]:

$$C_3 : w_1(\mathbf{r}) \to \omega w_1(C_3 \mathbf{r}), \qquad\qquad C_3 : w_4(\mathbf{r}) \to \omega w_4(C_3 \mathbf{r}),$$
$$C_3 : w_2(\mathbf{r}) \to \bar{\omega} w_2(C_3 \mathbf{r}), \qquad\qquad C_3 : w_3(\mathbf{r}) \to \bar{\omega} w_3(C_3 \mathbf{r}),$$
$$C_2' : w_1(\mathbf{r}) \to w_3(C_2' \mathbf{r}), \qquad\qquad C_2' : w_2(\mathbf{r}) \to w_4(C_2' \mathbf{r}),$$

where $\omega = e^{2\pi i/3}$ and $\bar{\omega} = e^{-2\pi i/3}$. In other words, the orbitals $w_1$ and $w_3$ transform between themselves, and so do $w_2$ and $w_4$. The model also has time-reversal symmetry (TRS), under which $w_1 \leftrightarrow w_2$ and $w_3 \leftrightarrow w_4$.

Possible superconducting pairings are either singlet or triplet (there is no spin orbit coupling). It is reasonable to assume that pairing will be more important between sites that also correspond to the most important hopping integrals. Let us therefore concentrate on pairing states involving nearest neighbors on a given layer, i.e., between orbitals $w_1$ and $w_4$ (or $w_2$ and $w_3$). Because of the strong local repulsion in our model, we ignore on-site pairing. Let us then define the pairing operators

$$\begin{aligned}
S_{i,\mathbf{r}} &= c_{\mathbf{r},\uparrow} c_{\mathbf{r}+\mathbf{a}_i,\downarrow} - c_{\mathbf{r},\downarrow} c_{\mathbf{r}+\mathbf{a}_i,\uparrow}, & \text{(singlet)} \\
T_{i,\mathbf{r}} &= c_{\mathbf{r},\uparrow} c_{\mathbf{r}+\mathbf{a}_i,\downarrow} + c_{\mathbf{r},\downarrow} c_{\mathbf{r}+\mathbf{a}_i,\uparrow}, & \text{(triplet)}
\end{aligned} \qquad (1)$$

where $c_{\mathbf{r},\sigma}$ annihilates an electron at graphene site $\mathbf{r}$ of the first layer (in orbital $w_1$ or $w_4$ depending on the sublattice). The elementary vectors $\mathbf{a}_i$ are defined on Fig. 1, but apply to the layer in the current context. Likewise, we define operators $S'_{i,\mathbf{r}}$ and $T'_{i,\mathbf{r}}$ on the second layer, in terms of orbitals $w_2$ and $w_3$). Under the transformations $C_3$ and $C_2'$, the six singlet (triplet) pairing operators transform amongst themselves and may be organized into irreducible representations of $D_3$, as listed on Table 2. To make this table more concise, we have defined the

Table 2: Irreducible representations (irreps) of $D_3$ associated with the six pairing operators defined on nearest-neighbor sites, as defined in Eqs (2). (Un)primed operators belong to the second (first) layer.

| Irrep | singlet pairing | triplet pairing |
|---|---|---|
| $A_1$ | $(d+id)+(d'-id')$ | $(p+ip)-(p'-ip')$ |
| $A_2$ | $(d+id)-(d'-id')$ | $(p+ip)+(p'-ip')$ |
| $E$ | $[d-id \ , \ d'+id']$ | $[p-ip \ , \ p'+ip']$ |
| | $[s,s']$ | $[f,f']$ |

following combinations:

$$s = \sum_{\mathbf{r}} \left( S_{1,\mathbf{r}} + S_{2,\mathbf{r}} + S_{3,\mathbf{r}} \right) \tag{2a}$$

$$d+id = \sum_{\mathbf{r}} \left( S_{1,\mathbf{r}} + \omega S_{2,\mathbf{r}} + \bar{\omega} S_{3,\mathbf{r}} \right) \tag{2b}$$

$$d-id = \sum_{\mathbf{r}} \left( S_{1,\mathbf{r}} + \bar{\omega} S_{2,\mathbf{r}} + \omega S_{3,\mathbf{r}} \right) \tag{2c}$$

$$f = \sum_{\mathbf{r}} \left( T_{1,\mathbf{r}} + T_{2,\mathbf{r}} + T_{3,\mathbf{r}} \right) \tag{2d}$$

$$p+ip = \sum_{\mathbf{r}} \left( T_{1,\mathbf{r}} + \omega T_{2,\mathbf{r}} + \bar{\omega} T_{3,\mathbf{r}} \right) \tag{2e}$$

$$p-ip = \sum_{\mathbf{r}} \left( T_{1,\mathbf{r}} + \bar{\omega} T_{2,\mathbf{r}} + \omega T_{3,\mathbf{r}} \right) \tag{2f}$$

and likewise for the combinations $s'$, $d' \pm id'$, etc. for the second layer. A similar analysis could be carried out with longer-range pairing, with the same classification: This would simply add harmonics to the basic pairing functions.

This organization into representations of $D_3$ is contingent on the importance of the inter-layer hopping $t_{13}$, which is an order of magnitude smaller than the intra-layer hopping. If $t_{13}$ were zero, the two layers would be independent, the symmetry would be upgraded to $C_{6v}$ and the classification of pairing states would be the same as in Ref. [27], with representations $A_1$ ($s$), $A_2$ ($f$), $E_1$ ($p \pm ip$) and $E_2$ ($d \pm id$). Since $t_{13}$ is small, we expect that the different pairing states of Table 2 (for a given total spin) will be nearly impossible to differentiate from an energetics point of view, except for the difference between $s$ and $d \pm id$ (or between $f$ and $p \pm ip$).

# 3 Cluster dynamical mean field theory

In order to probe the possible existence of superconductivity in this model, we use cluster dynamical mean-field theory (CDMFT) [28–30] with an exact diagonalization solver at zero temperature (or ED-CDMFT). Let us summarize this method.

## 3.1 General description

The infinite lattice is tiled into identical, repeated units; this defines a superlattice, and an associated reduced Brillouin zone, smaller than the original Brillouin zone. In the present study

the unit cell of the superlattice (or *supercell*) is made of four clusters of four sites each: Two clusters tile each of the two layers (Fig. 2c). Ref. [31] explains the particulars of CDMFT when the supercell contains more than one cluster. Each cluster is coupled to a bath of uncorrelated, auxiliary orbitals, and is governed by an Anderson impurity model (AIM):

$$H_{\text{imp}} = H_c + \sum_{i,r} \theta_{ir} \left( c_i^\dagger a_r + \text{H.c.} \right) + \sum_r \epsilon_r a_r^\dagger a_r \,, \tag{3}$$

where $H_c$ is the infinite-lattice Hamiltonian, but restricted to the cluster, $c_i$ annihilates an electron on orbital $i$ of the cluster ($i$ labels both site and spin) and $a_r$ annihilates an electron on orbital $r$ of the bath. The bath parameters ($\epsilon_r$, $\theta_{ir}$) are found by imposing a self-consistency condition, as explained below.

Hamiltonian (3) is solved by exact diagonalization. Without taking into account any symmetry of the Hamiltonian, the dimension of the Hilbert space for an impurity of 4 cluster sites and 6 bath sites would be $d = 4^{4+6} \sim 10^6$. Because we are investigating a broken symmetry state where the number of particles is not conserved, the only Abelian symmetry that can be used is the conservation of the $z$-component of the spin (we cannot use point group symmetries in general). Assuming a $S_z = 0$ state (singlet or triplet), this reduces the dimension of the Hilbert space to 184,756.

The electron Green function on the cluster, $\mathbf{G}_c(\omega)$, is needed by CDMFT. We use the band Lanczos method to compute it; for details, please see Refs [32,33]. This method provides a Lehmann representation for the Green function. This is a $L_c \times L_c$ matrix, $L_c$ being the number of orbitals on the cluster (including spin). It may be expressed in terms of the electron self-energy on cluster $c$, $\mathbf{\Sigma}_c(\omega)$, and the associated hybridization function $\mathbf{\Gamma}_c(\omega)$:

$$\mathbf{G}_c(\omega)^{-1} = \omega - \mathbf{t}_c - \mathbf{\Gamma}_c(\omega) - \mathbf{\Sigma}_c(\omega) \,, \tag{4}$$

where

$$\Gamma_{c,ij}(\omega) = \sum_r \frac{\theta_{ir} \theta_{jr}^*}{\omega - \epsilon_r} \tag{5}$$

and $\mathbf{t}_c$ is the matrix of one-body terms of $H_c$ (including the chemical potential $\mu$).

The fundamental approximation of CDMFT is to replace the exact electron self-energy by the self-energy obtained by assembling the various cluster self-energies:

$$\mathbf{\Sigma}(\omega) = \bigoplus_c \mathbf{\Sigma}_c(\omega) \,, \tag{6}$$

where the direct sum is carried over the various clusters forming the supercell. The Green function on the infinite lattice is then approximated by

$$\mathbf{G}(\tilde{\mathbf{k}}, \omega) = \left[ \omega - \mathbf{t}(\tilde{\mathbf{k}}) - \mathbf{\Sigma}(\omega) \right]^{-1} \,, \tag{7}$$

where $\tilde{\mathbf{k}}$ is a wave vector in the reduced Brillouin zone and $\mathbf{t}(\tilde{\mathbf{k}})$ is the noninteracting dispersion relation expressed in real space within the supercell and in reciprocal space within the reduced Brillouin zone. If $L_{\text{tot}} = \sum_c L_c$ is the total number of orbitals in the supercell, then $\mathbf{G}(\tilde{\mathbf{k}}, \omega)$, $\mathbf{t}(\tilde{\mathbf{k}})$ and $\mathbf{\Sigma}(\omega)$ are $L_{\text{tot}} \times L_{\text{tot}}$ matrices. We further define the projected Green function

$$\bar{\mathbf{G}}(\omega) = \int \frac{d^2\tilde{k}}{(2\pi)^2} \mathbf{G}(\tilde{\mathbf{k}}, \omega) \,. \tag{8}$$

This is the Fourier transform of the infinite-lattice Green function (7) to a single supercell around the origin. The CDMFT self-consistency condition requires that the $L_c \times L_c$ diagonal blocks of $\bar{\mathbf{G}}(\omega)$ (noted $\bar{\mathbf{G}}_c(\omega)$) should coincide with the corresponding cluster Green functions

$\mathbf{G}_c(\omega)$. This cannot be satisfied exactly with a finite number of bath orbitals, because it should hold for all frequencies and only a finite number of bath parameters are at hand. Therefore this condition is replaced by the optimization of a distance function:

$$d(\boldsymbol{\epsilon}, \boldsymbol{\theta}) = \sum_{c, i\omega_n} W(i\omega_n) \left[ \mathbf{G}_c(i\omega_n)^{-1} - \bar{\mathbf{G}}_c(i\omega_n)^{-1} \right], \tag{9}$$

where the weights $W(i\omega_n)$ are chosen in some appropriate way along a grid a Matsubara frequencies associated with some fictitious temperature $\beta^{-1}$. This is where some arbitrariness arises in the method, as will be commented below.

Let us then quickly summarize the actual CDMFT algorithm:

1. A trial value of the bath parameters $(\epsilon_r, \theta_{ir})$ is chosen. When looping over an external parameter, the previous converged value or an extrapolation thereof is chosen.

2. The cluster Green functions $\mathbf{G}_c(\omega)$ are computed, with the help of an *impurity solver* (here an exact diagonalization method).

3. The projected Green functions $\bar{\mathbf{G}}_c(\omega)$ are computed from Eqs (4), (7) and (8).

4. A new set of bath parameters is found by minimizing the distance function (9) with respect to the bath parameters entering $\mathbf{G}_c(\omega)$ through Eqs (4,5) for a fixed value of $\boldsymbol{\Sigma}_c$.

5. We go back to step 2 until the bath parameters or the hybridization functions $\boldsymbol{\Gamma}_c$ converge.

Once the converged solution is found, various quantities may be computed either from the impurity model ground state (averages, etc.) or from the associated lattice Green function $\mathbf{G}(\tilde{\mathbf{k}}, \omega)$.

## 3.2 Cluster-bath system

The cluster-bath system for the current problem is illustrated on Fig. 2. The supercell contains four 4-site clusters; one layer is illustrated on Panel (c). Note that the only hopping term included in the impurity model is $t_{14}[0,0]$ and its equivalents, represented by red lines on Fig. 1. The other hopping terms have an effect through the self-consistent CDMFT procedure.

Each cluster contains four sites and six bath orbitals and the various bath parameters are illustrated on panels (a) and (b). The four black, numbered circles are the cluster sites *per se*. The six red squares are the bath orbitals. Even though their positions have no meaning, they are, on this diagram, assigned a virtual position that makes them look as if they were physical sites on neighboring clusters. They are then given "nearest-neighbor" hybridizations $\theta_{1,2}$ and "second-neighbor" hybridizations $\eta_{1,2}$. In order to probe superconductivity, we add pairing amplitudes within the bath itself, as shown on Fig. 2b: Two pairing amplitudes $d_{1,2}$ between consecutive bath orbitals, and two others $p_{1,2}$ between "second neighbor" bath orbitals. In the context of Eq. (3), these pairing amplitudes must be understood in the restricted Nambu formalism, in which a particle-hole transformation is applied to the spin-down orbitals, giving the pairing operators the looks of hopping amplitudes. Specifically, in terms of the multiplet $(C_\uparrow, C_\downarrow^\dagger, A_\uparrow, A_\downarrow^\dagger)$, where $C_\sigma = (c_{1,\sigma}, c_{2,\sigma}, c_{3,\sigma}, c_{4,\sigma})$ and $A_\sigma = (a_{1,\sigma}, \cdots, a_{6,\sigma})$ ($\sigma = \uparrow, \downarrow$), the non-interacting part of the impurity Hamiltonian takes the form

$$H_{\text{imp}}^0 = \begin{pmatrix} C_\uparrow^\dagger & C_\downarrow & A_\uparrow^\dagger & A_\downarrow \end{pmatrix} \begin{pmatrix} \mathbf{T} & \boldsymbol{\Theta} \\ \boldsymbol{\Theta}^\dagger & \mathbf{E} \end{pmatrix} \begin{pmatrix} C_\uparrow \\ C_\downarrow^\dagger \\ A_\uparrow \\ A_\downarrow^\dagger \end{pmatrix}, \tag{10}$$

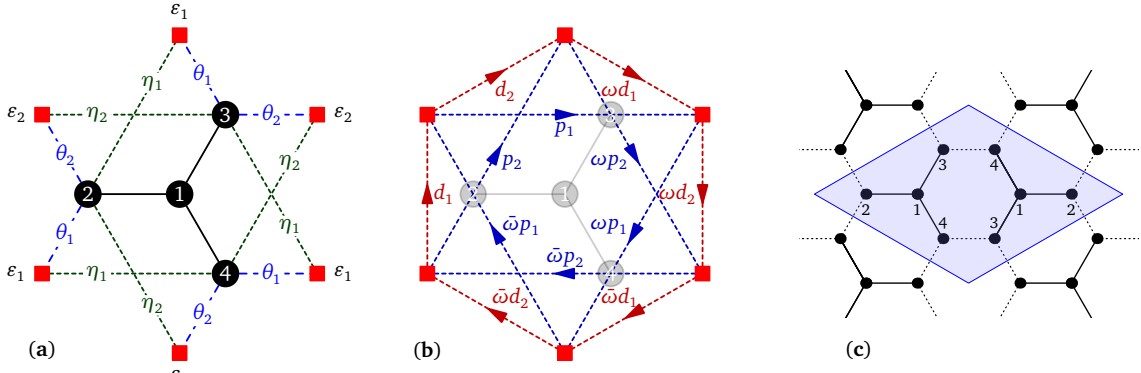

Figure 2: Schematic representation of the impurity model used in this work. Each cluster is made of four lattice sites (numbered black dots) and six bath orbitals (red squares). The normal-state bath parameters are shown on Panel (a): Two different bath energies $\varepsilon_{1,2}$, four different hybridizations $\theta_{1,2}$ and $\eta_{1,2}$. The anomalous bath parameters are shown on Panel (b). As shown, they are optimized for studying the $p + ip$ state: Two complex-valued triplet pairings $d_{1,2}$ between "nearest-neighbor" bath orbitals, and two other complex-valued triplet pairings $p_{1,2}$ between "second-neighbor" bath orbitals, all modulated by powers of the complex amplitude $\omega = e^{2\pi i/3}$ as one goes around ($\bar{\omega} = \omega^2 = \omega^{-1}$). The unit cell of the impurity model contains four copies of this cluster: Two on the bottom level ($w_{1,3}$), two on the top level ($w_{2,4}$). On each level, they are arranged as shown on Panel (c) (the 4-site cluster on the right is the inversion of the one on the left, and the bath parameters are the same on the two clusters, except for the sign of the triplet pairings, which are inverted).

where

$$\mathbf{T} = \begin{pmatrix} \mathbf{t}_c & 0 \\ 0 & -\mathbf{t}_c \end{pmatrix}, \qquad \mathbf{\Theta} = \begin{pmatrix} \boldsymbol{\theta} & 0 \\ 0 & -\boldsymbol{\theta}^* \end{pmatrix}, \qquad \mathbf{E} = \begin{pmatrix} \boldsymbol{\epsilon} & \boldsymbol{\Delta}^\dagger \\ \boldsymbol{\Delta} & -\boldsymbol{\epsilon} \end{pmatrix}. \tag{11}$$

Here $\mathbf{t}_c$ is the hopping matrix restricted to the cluster, $\boldsymbol{\theta}$ is a $4 \times 4$ matrix containing the parameters $\theta_{1,2}$ and $\eta_{1,2}$, $\boldsymbol{\epsilon}$ is a diagonal matrix containing the bath energies $\varepsilon_{1,2}$ and $\boldsymbol{\Delta}$ is a $6 \times 6$ matrix containing the parameters $d_{1,2}$ and $p_{1,2}$.

In total, the AIM contains 10 bath parameters, some real, some complex. The impurity Hamiltonian does not contain pairing operators on the cluster sites themselves. However, the operators defined in Eqs (1) may develop a nonzero expectation value on the impurity through the self-consistent bath.

The hybridization pattern shown in the figure is appropriate for triplet pairing (it is directional, as indicated by the arrows) in a $p + ip$ state (because of the phases $\omega$ and $\omega^2 = \bar{\omega}$ appearing in the bath pairing amplitudes as one circles around). This may be readily adapted to probing a $p - ip$ state (by replacing $\omega \leftrightarrow \bar{\omega}$) or a $f$ state (by replacing $\omega, \bar{\omega} \to 1$). Likewise, singlet states are probed by introducing singlet pairing between bath sites. In principle, we could leave all pairings free, at the price of tripling the number of bath parameters, but CDMFT convergence has proven problematic when this was tested. It is easier, and no less general, to separately probe the $p \pm ip$ and $f$ states (and likewise for the singlet states).

One could also treat the bath parameters of all four clusters of the supercell as independent. In practice, this is not necessary as they are related. The two clusters belonging to the same layer have identical bath parameters by symmetry, except for the triplet pairings which must change sign between the two clusters because the second cluster is obtained from the first

by a spatial inversion. According to Table 2, we expect the complex-valued bath parameters of the second layer to be the complex conjugates of those of the first layer. These constraints effectively reduce the total number of variational parameters to the equivalent of 13 real parameters.

Minimizing the distance function (9) is done by the Nelder-Mead or the conjugate-gradient method as implemented in SciPy. These methods do not guarantee a global minimum, but only a local one. Because of this, jumps in the bath parameters might occur as a function of an external (control) parameter like the chemical potential $\mu$, and we would expect that this manifests itself as a hysteresis when cycling over $\mu$. We have observed no such hysteresis in the present study. This being said, the CDMFT algorithm summarized above contains an iteration over impurity models that defines a very complex nonlinear system that rather complicates this simple expectation. Failure to converge often manifests itself by oscillations between two or more sets of bath parameters and experience shows that increasing the parameter set does not necessarily alleviate this problem.

## 4 Results and discussion

We have probed the different states listed in Table 2 using the above CDMFT setup. In order to reach a solution from scratch, we have used the following staged approach: (i) Owing to the small value of $t_{13}$, a one-layer model was first studied. (ii) An external field of each of types (2) was then applied to the cluster in order to induce a nonzero average pairing forcefully. This external field was then reduced to zero in a few steps, each time starting from the previous solution. (iii) Once a nontrivial solution was found in this way at zero external field, the second-layer was added (with a complex conjugated bath system, e.g., $p-ip$ instead of $p+ip$). (iv) the solution found was then scanned as a function of chemical potential within the two-layer model. The most delicate step is to find a first solution; scanning over parameters of the model (such as the chemical potential or the interaction) is easier since the solution at a given set of model parameters provides an initial trial solution for the next parameter set. Computing time varies depending on convergence rate, but is typically of the order of 10 minutes per parameter set once the scan is in motion, with code highly optimized for speed; memory needs are relatively modest at 3-4 gigabytes.[1]

We found a nonzero solution for $p \pm ip$ pairing extending over a wide range of doping. Fig. 3 shows the average $p+ip$ order parameter on a cluster of the first layer, as a function of electron density on the cluster, for a local repulsion $U = 2$ meV. The order parameter is the ground-state expectation value of operator (2e) restricted to the cluster within the impurity model. Several variants of the CDMFT procedure are illustrated, which we must now explain. The distance function (9) depend on a set of weights $W(i\omega_n)$ and a fictitious temperature $\beta^{-1}$. The values of $\beta$ (in meV$^{-1}$) are indicated in the legend of Fig. 3. The grid of Matsubara frequencies then stops at some cutoff value taken to be $\omega_c = 2$ meV in this work. The curve labeled $\beta = 50$ (blue dots) is obtained by setting all weights to the same value. The other curves (with a $\Sigma$ label) are obtained by setting the weights proportional to the self-energy $|\Sigma(i\omega_n)|$ (the norm of the matrix). This is justified if one considers DMFT from the point of view of the Potthoff functional [34, 35]. In particular, it gives more importance to very low frequencies in an insulating state, as the self-energy then grows as $\omega \to 0$. We expect the superconducting order parameter to be minimum, if not zero, at quarter ($n = 0.5$) or three-quarter ($n = 1.5$) filling, as observed in experiments. Indeed, this commensurate filling leads

---

[1]Adding just a few orbitals to the impurity problem would dramatically increase the resources needed: Going from 6 to 9 bath orbitals, for a total of 13 orbitals in the impurity model, would increase the Hilbert space dimension 50-fold, with a corresponding increase in memory usage and an even sharper increase in computing time.

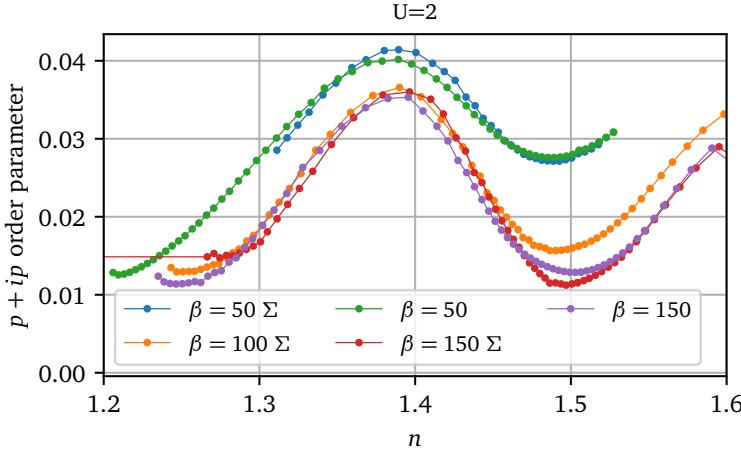

Figure 3: $p + ip$ order parameter found by CDMFT, as a function of electron density $n$, for $U = 2$ meV and several variants of the CDMFT procedure explained in the text. Only the electron-doped results ($n > 1$) are shown for clarity.

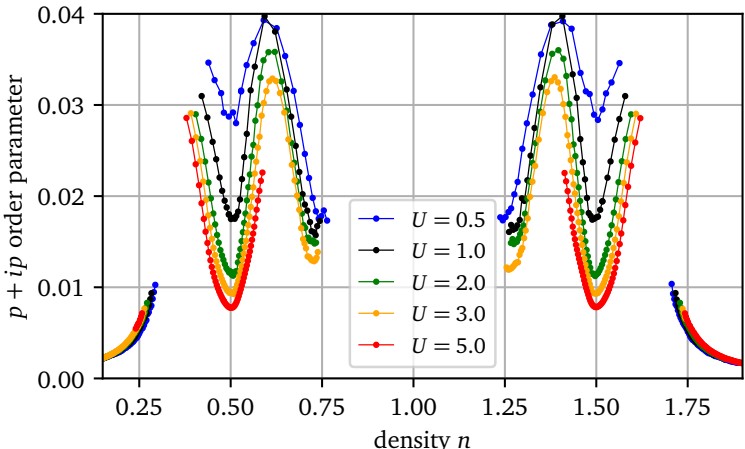

Figure 4: $p + ip$ order parameter found by CDMFT, as a function of electron density $n$, for several values of Hubbard $U$ (in meV). The order parameter is the ground state average of the operator (2e), restricted to the cluster. The density $n$ is the ground-state average occupation of the cluster. One of the clusters of the first layer was used for these averages. Clusters on the second layer would show the opposite chirality ($p - ip$).

to an insulating state at the magic angle 1.08° [5] and superconductivity occurs on either side of this filling value. We see that this is not exactly the case in the data sets of Fig. 3, although using a higher $\beta$ and, to a lesser extent, a self-energy modulated set of weights, greatly helps. We will stick to the value $\beta = 150$ and use a self-energy modulated set of weights in what follows.

Figure 4 shows the $p + ip$ order parameter as a function of electron density for the full range of solutions obtained, and five values of the one-site repulsion $U$ (in meV). We note that the system is almost (but not exactly) particle-hole symmetric. Superconductivity is strongly suppressed near half-filling (CDMFT ceases to converge to a superconducting solution when $|n - 1| \lesssim 0.2$). Superconductivity is partially suppressed at quarter- and three-quarter filling

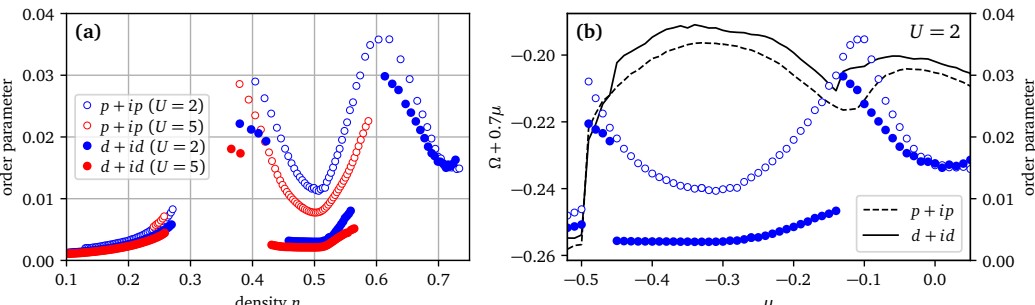

Figure 5: Left panel: $d + id$ order parameter found by CDMFT (filled circles), as a function of electron density $n$, compared with the $p + ip$ order parameter (open circles), for $U = 2$ meV and $U = 5$ meV. The $d + id$ order parameter is the ground state average of the operator (2b), restricted to the cluster. Again, clusters on the second layer would show the opposite chirality ($d - id$). Right panel: For $U = 2$ and as a function of chemical potential $\mu$, the same chiral order parameters as in the left panel, as well as the value of the Potthoff functional $\Omega$ for each solution. The $p + ip$ solution (dashed curve) has a lower energy than the $d + id$ solution (full curve). A multiple of $\mu$ was added to $\Omega$ to rectify the curves and improve clarity.

($n = 0.5, 1.5$) and this suppression increases with $U$. Despite a strong suppression of superconductivity at $n = 0.5$ and $n = 1.5$, a Mott state is not fully obtained there for the range of $U$ studied. This is likely caused by our neglect of extended interactions. Note the gap in the solutions in the vicinity of $n = 0.3$ and $n = 1.7$; the solutions exist for all values of chemical potential $\mu$ around these values, but a discontinuity leads to the forbidden regions when plotted as a function of density.

We also found a weaker singlet solution with $d + id$ symmetry, as illustrated on Fig. 5a for $U = 2$ meV and $U = 5$ meV. The singlet solution has a smaller order parameter than the triplet solution, especially in the vicinity of $n = 0.5$ and $n = 1.5$, where it is strongly suppressed and suffers from a discontinuity (we only show the hole-doped case for clarity). A possible way to discriminate between the triplet and singlet solutions is to compare the energies of each. An optimal way to estimate the energy in CDMFT is to borrow the expression of the Potthoff self-energy functional from the variational cluster approximation [36, 37], as explained in Ref. [38]. The expression of the Potthoff functional is

$$\Omega = E_0 + \text{Tr} \ln[-(\mathbf{G}_0^{-1} - \mathbf{\Sigma})^{-1}] - \text{Tr} \ln(-\mathbf{G}_c), \tag{12}$$

where $E_0$ is the ground state energy per site of the impurity model (including the chemical potential contribution), and the functional trace Tr represents an integral over frequencies and wave vector. It is an approximation to the grand potential $\Omega = E - \mu N$ of the system at zero temperature, given that the CDMFT is not far from the solution to Potthoff's variational principle [36]. Figure Fig. 5b shows the Potthoff functional of the two solutions ($p + ip$ and $d + id$) at the same time as the corresponding order parameters, as a function of chemical potential $\mu$. The grand potential of the triplet is consistently lower than that of the singlet, except for an isolated point near a discontinuity. We have also compared directly the ground state energies $E_0$ of the corresponding two impurity models, and the same conclusion holds: the singlet $d + id$ solution has a higher energy, a smaller order parameter, and is thus subdominant.

We were not able to resolve the different representations of $D_3$, as listed on Table 2. In other words, the energy difference between the $A_1$, $A_2$ and $E$ representations is likely too small to have an effect on the CDMFT convergence procedure. This is due to the small value of the inter-layer hopping $t_{13}$. It is however important to assign opposite chiralities to the two layers.

The effective model used was based on the parameters of Ref. [1], appropriate for a twist angle $\theta = 1.30°$. Would our conclusions change for different, small twist angles, such as the ones found in Ref. [6] ($\theta = 1.05°$, $1.16°$)? Maybe. But a similar CDMFT of the nearest-neighbor Hubbard model on the graphene lattice has shown triplet pairing to be dominant [27]; so did a RPA study of bi-layer silicene [39], which is likewise based on the graphene lattice.

Let us compare our conclusions with some other works having found superconductivity in effective models for twisted bilyaer graphene. Ref. [9] finds triplet superconductivity as a Kohn-Luttinger instability, but is essentially a weak-coupling analysis, contrary to ours. Ref. [13] finds triplet superconductivity near $n = 0.5$, but with $f$ symmetry, using a numerical renormalization group approach expected to be valid from weak to moderate coupling. Our strong-coupling calculations could not stabilize $f$-wave superconductivity. Kennes *et al.* [8] find $d + id$ superconductivity near $n = 1$ using a renormalization-group approach followed by an mean-field analysis. Zhang *et al.* [14] arrive at the same conclusion, using constrained path Monte Carlo, and so do Chen et al [17]. These three works do not contradict ours, since our prediction concerns mostly regions around $n = 0.5$ and $n = 1.5$, not $n = 1$.

A possible improvement to the present study would be to include extended interactions, for example derived from an on-site Coulomb interaction at the AA sites [23, 24]. We expect that including such interactions would hinder pairing at quarter filling. This would require adding inter-orbital interactions $U_{1,2}$ ($U_{3,4}$) between orbitals $w_1$ and $w_2$ ($w_3$ and $w_4$). Unfortunately, since orbitals $w_1$ and $w_2$ belong to different clusters in our CDMFT setup, this cannot be implemented as is. The effect could be studied within a different quantum cluster approach, such as the variational cluster approximation [27, 34, 40], which in practice allows larger clusters. Alternately, inter-cluster interaction terms could be treated at the mean field level, as done, for instance, in Refs. [27, 41]. Interactions that do not have a density-density form (and thus not diagonal in the Wannier basis) would, naturally, complicate matters.

A legitimate question is whether other broken symmetries could compete with superconductivity in the phase diagram. We expect charge order to be a serious contender at commensurate filling (in particular $n = 0.5$ and $n = 1.5$), provided extended interactions are taken into account. It is possible that the superconducting order that we found would disappear precisely at these fillings, either because of the extended interactions or out of competition with charge order. Likewise, antiferromagnetism is likely to appear at half-filling ($n = 1$), where superconductivity is suppressed, because of a suppression of the density of states related to Mott physics. Again we leave this question for future work.

**Funding information** DS acknowledges support by the Natural Sciences and Engineering Research Council of Canada (NSERC) under grant RGPIN-2015-05598. Computational resources were provided by Compute Canada and Calcul Québec.

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
