# Peer review of "Chiral $p$-wave superconductivity in twisted bilayer graphene from dynamical mean field theory"

_SciPost Physics, doi:SciPost Phys. 11, 017 (2021)_

## Round 3 · Referee Report · Anonymous (Referee 1) · 2021-6-23

Report

The authors have extended the manuscript and included the discussions requested by the referees.
This manuscript is well written and presents interesting results on twisted bilayer Graphene.
I support publication in SciPost Physics.

---

## Round 3 · Referee Report · Anonymous (Referee 3) · 2021-6-24

Report

I find that the authors answer my remarks/questions in a satisfactory manner, in particular the extended new version of the conclusion stress the importance of that work in relation with already published studies. It seems to me that they also answer properly the questions of the other referees. So I think this manuscript deserves to be published in SciPost Physics.

---

## Round 3 · Referee Report · Anonymous (Referee 2) · 2021-7-6

Report

In their revised manuscript the authors answered to all pertinent questions and remarks of the referees. In particular, the method section has been extended and includes the technical points requested. The discussion/conclusion has been improved, too, and now not only presents the results in context of existing literature, but also clearly states open questions.
The manuscript is well written and properly formatted.
Overall, I recommend publication in SciPost Physics.

---

## Round 3 · Author Response

We will address comments by each referee in turn.

Response to first Referee report

1. The main issue raised here is the apparent contradiction between the use of a four-band tight-binding model proposed in Phys. Rev. X 8 031088 and a argument given in Phys. Rev. B 99, 195455 that a four-band model cannot sustain the fragile topology expected in the low-energy model. Please see the discussion to that effect in Phys. Rev. B 102, 075142. While it is true that using localized Wannier functions with a purely local action of the symmetries is strictly incompatible in the continuum limit with the fragile topology, the error committed doing this is small. In any case, this issue is dwarfed by the approximations we make by keeping only the dominant hopping terms. In addition, we expect that introducing strong interactions would destroy any fragile topology that would be preserved in a noninteracting model. Hence, for the purpose of identifying the most favored superconducting pairing in a strongly interacting model, we believe this should not be an issue.

  1. A concern is the use of local interactions when in reality we expect the interaction to extend over neighboring Wannier states. This a very legitimate concern and is the main weakness of our work. It is rooted in the small size of the cluster used in our CDMFT computations. We could keep extended interactions contained in a cluster, but this would ignore inter-cluster extended interactions. A way to alleviate this problem would be to implement a form of inter-cluster mean-field theory. We mention this possibility in the revised manuscript, but reserve its implementation for future work.

  2. The referee asks for some precisions about the methodology: the exact diagonalization, the spectral function, etc. We provide some of this in the revised version.

  3. The referee wonders why we expect the order parameter to have a minimum at the quarter filling (n=0.5). We expect this because (i) this is observed in experiments at magic angle 1.08 degrees and is the consequence of an insulating state at that commensurate filling value. Commensurate filling is more likely to lead to an insulating state because of interactions, in particular extended interactions. In our study, even if extended interactions are not taken into account a sharp minimum in the superconducting order parameter is observed. Note that our revised results no longer use counterterms to enforce the precise location of this minimum: the additional hopping terms included suffice.

Response to second Referee report

We thank the referee for this very thorough review of our work.

1A. The referee suggests that we refine the tight-binding model by adding another set of hopping parameters that we neglected since we elected to keep the largest two sets of couplings on any given layer. We followed that suggestion and redid the computations including the next largest hopping amplitude. This explains in part why our response was delayed. The results are qualitatively the same for the triplet case but resurrect the singlet solution, which is now stabilized, although with a smaller order parameter and higher energy than the triplet solution. These additional hopping terms naturally lead to a minimum at fillings 0.5 and 1.5 without need for cluster counterterms, so that new technique was dropped from the paper.

1B. The referee asks us to justify the range of values of U used in the computations. We did not use any estimate from the references given by the referee to select our range of U (from 0.5 meV to 3 meV). We simply chose a range that would take us from intermediate coupling to strong coupling. In the revised version we situate values suggested by those references in that range. In particular, the value of U quoted by Cao et al (U~20 meV) seems very large in this context. We added the value U=5meV. This is near the lower bound of the estimates of Goodwin et al. Yet we are of the opinion that this value being already larger than the bandwidth of the model, no new physics will emerge by pushing U beyond this value.

2A. We bring precisions about the methods in the revised version, as suggested by the referee. However, we do not go as far as performing a new set of computations with 3 bath sites per boundary cluster site, which would bring the total number of orbitals to 13. This would take considerable resources given that (i) the cluster has no point group symmetry to be exploited easily (ii) complex numbers are needed, (iii) the exact diagonaliations have to be performed very many times. Also, our reasoning for the bath parameterization is inspired considering the bath sites as a sort of mock-up of the neighboring sites, which leads to this impurity model of 6 bath sites per cluster.

2B. We agree with the referee that these additional orders are interesting to study and we indeed plan on this. In the context of the present paper, this would take us a bit too far. We are pretty much convinced that such a model would lead to antiferromagnetism at half-filling and we are currently investigating, in a different project, the possibility of charge order near quarter filling. But we use slightly different methods for this and including these considerations in the present paper would unduly delay publication as well as break the unity of the paper in terms of methods and impurity model.

2C. We agree. It turns out that including one additional set of subleading hopping terms has stabilized the singlet solution (d+id) and we cover this in the revised version.

Response to third Referee report

1. The model we use does not exactly neglect hopping beyond first neighbors. In the version of the manuscript reviewed by the referees, hopping was kept with first and third neighbors. In the revised version we have added hopping with fourth neighbors. Third and fourth neighbor hopping are not present within the small four-site cluster but still have an effect through the self-consistency relation. To clarify this point, we have added a figure that sketches the hopping terms used in the paper. We also added a remark to the effect that hopping terms beyond nearest neighbor have an effect through the CDMFT self-consistency relation.

  1. The revised version of the manuscript comments on the relations of our results with other published results for the superconducting order parameter.

  2. The conclusion has been expanded.

---

## Round 3 · List of Changes

1. Comments/explanations have been added on a number of points: a) The relevance of the underlying tight-binding model given its incompatibility with the expected topological properties. b) Some technical points of the methodology required by the referees c) Our choice of the local interaction strength U d) Our neglect of extended interactions e) The conclusion/discussion has been expanded. In particular, a brief comparison is made with some other works predicting particular forms of superconductivity.
  2. Most important, the truncated set of hopping terms kept for the study now includes the next (fourth-neighbour) hoping terms, as suggested by the second referee. Computation were entirely redone because of this.
  3. These new computations show the existence of a subdominant singlet solution. We present this solution and explain why it is subdominant (i.e. of higher energy than the triplet solution).
  4. The CDMFT procedure no longer include a counterterm to force the uniformity of the density on the cluster, as it is no longer required to make the minimum of superconductivity coincide with the commensurate (0.5 and 1.5) filling.
  5. Overall, the paper has been expanded and one stand-alone figure has been added

---

## Editorial Decision

published